# Antithrombotic Effect of the Ethanol Extract of *Angelica gigas* Nakai (AGE 232)

**DOI:** 10.3390/life11090939

**Published:** 2021-09-09

**Authors:** Pia Loreto Werlinger Bravo, Hui Jin, Hyunwoo Park, Min Sang Kim, Hirofumi Matsui, Hanki Lee, Joo-Won Suh

**Affiliations:** 1Graduate School of Interdisciplinary Program of Biomodulation, Myongji University, Yongin 17058, Korea; piawerlinger@mju.ac.kr; 2Center for Nutraceutical and Pharmaceutical Materials, Myongji University, Yongin 17058, Korea; jh@kimc.or.kr (H.J.); hwpark75@snu.ac.kr (H.P.); kms0177@naver.com (M.S.K.); 3Health Park Co., Ltd., Seoul 06627, Korea; 4Graduate School of Medical and Dental Sciences, Kagoshima University, Kagoshima 890-8544, Japan; hmatsui@md.tsukuba.ac.jp; 5MJ Bioefficacy Research Center, Myongji University, Yongin 17058, Korea

**Keywords:** thrombus formation, acetylsalicylic acid, *Angelica gigas* Nakai, improvement of blood circulation, anti-platelet aggregation, bleeding in the gut, side effect

## Abstract

Cardiovascular diseases, such as stroke, are the most common causes of death in developed countries. Ischemic stroke accounts for 85% of the total cases and is caused by abnormal thrombus formation in the vessels, causing deficient blood and oxygen supply to the brain. Prophylactic treatments include the prevention of thrombus formation, of which the most used is acetylsalicylic acid (ASA); however, it is associated with a high incidence of side effects. *Angelica gigas* Nakai (AG) is a natural herb used to improve blood circulation via anti-platelet aggregation, one of the key processes involved in thrombus formation. We examined the antithrombotic effects of AGE 232, the ethanol extract of *A. gigas* Nakai. AGE 232 showed a significant reduction in death or paralysis in mice caused by collagen/epinephrine-induced thromboembolism in a dose-dependent manner and inhibition of collagen-induced human platelet aggregation in a concentration-dependent manner. Additionally, AGE 232-treated mice did not show severe bleeding in the gut compared to ASA-treated mice. AGE 232 resulted in a decrease in the number of neutrophils attached to the human umbilical vein endothelial cells (HUVECs) and lower inhibition of COX-1 in response to bleeding and damage to blood vessels, a major side effect of ASA. Therefore, AGE 232 can prevent thrombus formation and stroke.

## 1. Introduction

Thrombosis is the localized clotting of blood, which affects blood circulation and may cause stroke. Currently, the following three types of strokes have been described: subarachnoid hemorrhage, intracerebral hemorrhage and ischemic stroke (3%, 10% and 87% of the total cases, respectively) [1]. Around 3.2 million people were reported to suffer from stroke, which was found to be the second leading cause of death in 2015. The prevalence of this disease is estimated to increase by approximately 10% in the next 20 years, and 116 million people may have one or more forms of cardiovascular disease, costing around 818.1 billion USD by 2030 [2].

Ischemic strokes start with atherothrombosis, which is characterized by atherosclerotic plaque disruption, leading to platelet activation and thrombus formation, which is released into the bloodstream, blocking the blood flow in the artery and causing ischemia to the target organ [3]. Platelets play a central role in thrombus formation. Platelets are enucleated blood cells originating from the cytoplasm of megakaryocytes in the bone marrow and circulate in the bloodstream and check the integrity of the vascular system, identifying the normal endothelial cell lining and areas with lesions [1]. Platelet adhesion, activation and aggregation at the sides of vascular endothelial disruption caused by atherosclerosis are key events in arterial thrombus formation [4].

Antiplatelet drugs have served as cornerstones in the prevention and treatment of cardiovascular diseases [5]. Acetylsalicylic acid (ASA) is the main component of aspirin, which is the most used antiplatelet drug for more than 50 years [3]. Aspirin has been used as a prophylactic treatment against stroke because of its inhibitory effect on the synthesis of platelets [6]; however, this drug also is associated with a high rate of side effects such as gastric lesions, ulcerations and erosions of the stomach. Aspirin is commonly used to treat pain, inflammation and fever and acts by inhibiting cyclooxygenase (COX) activity [7]. After oral administration, aspirin passes through the plasma membrane in a unionized form and affects gastric epithelial function and locally inhibits COX, reducing surface hydrophobicity, bicarbonate secretion, blood flow and endogenous mucosal defense [8].

Many traditional herbal medicines have been used in clinical practice because of their efficacy, safety profile and low toxicity [9]. *Angelica gigas* Nakai (AG) is an herbal medicine used for the treatment of dysmenorrhea, amenorrhea, abdominal pain [10] and various circulatory disorders, and demonstrates anti-inflammatory [11], and neuroprotective effects in ischemic stroke [12]. AG has been demonstrated to improve blood circulation via anti-platelet aggregation [13]. In the present study, we evaluated the antithrombotic effect of AGE 232, the ethanol extract of *A. gigas* Nakai and decursin, and its mechanism as a natural alternative to prevent thrombus formation in the prevention of ischemic stroke.

## 2. Materials and Methods

### 2.1. Preparation of AGE 232

Preparation of AGE 232 was first carried out by extraction from dried *A. gigas* Nakai with 50% (*v*/*w*) ethanol corresponding to 10 times the weight of dried *A. gigas* Nakai for 3 h at 70 °C. This step was carried out 2 times. After filtration of the ethanol extract with a sieve (120 mesh), the filtrate was concentrated using a rotary evaporator till the concentration reached up to 10 °Brix. It was then sterilized at 85 ± 5 °C for 1 h. Finally, lyophilized AGE232 was prepared after filtration of the concentrate through a sieve (100 mesh).

### 2.2. Animals

Male 5-week-old ICR mice were used for the thromboembolism mouse model and the ASA-induced ulcerogenesis mice model. All mice were obtained from Raon Bio (Gyeonggi, South Korea) and housed as a group of four mice per cage in a room with a 12-h light cycle (lights on at 6:00 AM and off at 6:00 PM) at 23 ± 5 °C and 55 ± 5% humidity for 1 week before the experiment. The mice were housed in a pathogen-free environment with free access to water and a commercial pellet diet obtained from Raon Bio (Gyeonggi, South Korea).

All protocols for animal experiments were approved by the Institutional Animal Care and Committee (IACUC) of Myongji University (MJIACUC-2021002) and were conducted in accordance with the NIH Guide for the Care and Use of Laboratory Animals.

### 2.3. Thromboembolism Mouse Model

The antithrombotic activity of the ethanol extract of AGE 232, ASA and decursin was investigated in vivo by DiMinno’s method [14]. This model is characterized by massive activation of circulating platelets and the widespread formation of platelet thrombi in the microcirculation of the lungs, leading to disseminated pulmonary microembolism, and paralysis of the animals.

To assess sensorimotor coordination and balance performance, the mice were evaluated using the Rota Rod Unit. Mice were trained on the rotarod for 2 days before the experiment. On the day of the experiment, the animals were placed on the rotarod as the first trial for 5 min three times. After the first trial ended, a vehicle solution was orally administered to group 1 (control), aspirin (100 mg/kg), group 2 (positive control), and AGE 232 was orally administered at concentrations of 40, 80 and 160 mg/kg to groups 3, 4 and 5, respectively. Finally, a dose of 12.4 mg/kg of decursin was administered to group 6 to evaluate the antithrombotic effect of decursin, one of the major coumarins present in the AG extract. The time required for each animal to remain on the rotarod was recorded for three trials at a minimum interval of 5 min with a maximum trial length of 300 s each. The rotarod stopped automatically and recorded the time in 0.1 s till the mice fell on the floor. The speed was set to accelerate from 4 to 40 rpm until 300 s. The score was presented as the mean of three trials. After 1 h, 0.06 mL solution of epinephrine (1.7 μg/mL) and collagen (28.6 μg/mL) was injected into the tail vein to assess the thrombus. After 15 min, the mice were placed on the rotarod for three more trials. The experiment to record the number of deaths or paralyzed mice was conducted 3 times for 300 s each, and the percent protection was calculated using the following equation: Protection (%) = [1-(death or paralyzed mice) / total mice] × 100. The experiment was repeated four times.

### 2.4. ASA-Induced Ulcerogenesis Mouse Model, the Effect of Age 232 on the Gastric Tissue of Mice and Quantification of Ulceration

ASA-induced ulcerogenesis was performed following the method described by Szelenyu and Thiemer [15] with slight modifications. Briefly, male mice were separated into five groups (n = 4). Group 1 mice were separated as control and received only an oral administration of vehicle (10% DMSO in 0.5% carboxymethyl cellulose), groups 2 and 3 received an oral dose of ASA (30 and 300 mg/kg, respectively), and groups 4 and 5 received an oral dose of AGE 232 (30 and 300 mg/kg, respectively) once a day. The experiment was conducted for five consecutive days. On the fifth day, mice were sacrificed 4 h after the last ASA administration by the administration of 3% isoflurane for anesthesia induction and subsequent cervical dislocation. Immediately, the stomach was dissected, opened along the greater curvature, and stretched on cork plates. The inner surface was rinsed with PBS to remove any blood contaminants. The gastric mucosa was observed using a binocular stereomicroscope at 20X magnification (EZ4HD, Leica, Wetzlar, Germany). The stomach was examined for edema, necrosis and ulceration, and analyzed using ImageJ. The ulcer index was calculated using a modified Ganguly method [16], wherein the ulcer index was calculated as follows: Ulcer index (%) = [total ulcerated area/total mucosal area] × 100.

### 2.5. Cell Viability Using 3-(4,5-Dimethylthiazol-2-yl)-2-5-dyphenyltetrazolium Bromide (Mtt) Assay

To calculate the cell viability and toxicity of AGE 232 and ASA, RGM-1 cells were obtained from the Cell Engineering Division of RIKEN BioResource Center (RIKEN Cell Bank, Tsukuba, Ibaraki, Japan), seeded in 96-well plates and incubated for 24 h in 20% FBS (Gibco, Thermo Fisher Scientific, Inc., Waltham, MA, USA) DMEM/HamF12 Media (Sigma-Aldrich, St. Louis, MO, USA). After 24 h, cells were washed with warm PBS and treated with different concentrations of ASA (0.18, 0.9008, 1.8016, 3.6032, 9.008 and 36.032 μg/mL), AGE 232 (10, 50, 100, 150 and 500 μg/mL), vehicle (media with 1% DMSO) or only media for 24 h. Cell survival was measured using the MTT assay (Sigma-Aldrich, St. Louis, MO, USA). In brief, PBS containing 5 mg/mL MTT solution was diluted to a concentration of 0.5 mg/mL and incubated in a humidifier chamber containing CO_2_ for 4 h. The medium was aspirated from each well and 100 μL of DMSO was added to dissolve the formazan crystals. The absorbance of the sample in each well was measured using a microplate reader (TECAN Spectrofluor Plus, Maennedorf, Switzerland) at a wavelength of 570 nm. The half-maximal inhibitory concentration (IC_50_) of ASA and AGE 232 was calculated based on the percentage of remaining MTT radicals against the sample concentration using OriginPro software (ver 9.0, OriginLab Corp., Northampton, MA, USA).

### 2.6. Platelet Aggregation Assay

The inhibitory effects of AGE 232 and ASA were analyzed using an aggregometer (Chrono-Log, Havertown, PA, USA). Human platelet-rich plasma (PRP) was purchased from Gyeonggi Blood Center of the Korean Red Cross (Gyeonggi, South Korea) and diluted in PBS to a concentration of 2 × 10^8^ platelets/mL. ASA samples were prepared at concentrations of 2.5 mg/mL in 500 μL PRP. AGE 232 samples were prepared at final concentrations of 10, 20, 40, 80 and 160 mg/mL in 500 μL PRP, and all PRP samples contained 0.1% DMSO. The samples were incubated for 1 min with stirring at 1200 rpm. After incubation, 5 µg/mL of collagen reagent (Chrono-Log, Havertown, PA, USA) was added to the samples to induce platelet aggregation. Changes in light transmission were measured for 5 min after the induction of aggregation.

### 2.7. Endothelial Cell Leukocyte Adhesion Test

This experiment was performed to identify the glycoproteins present on the surface of leukocytes and endothelial cells responsible for the induction of adhesive interactions.

Human umbilical vein endothelial cells (HUVECs) were purchased from the American Type Culture Collection (ATCC^®^ CRL-1730™). HUVECs were collected by collagenase treatment. Cells were smeared on medium 199 containing 10% heat-inactivated bovine calf plasma, thymidine (2.4 mg/L), glutamine (230 mg/mL), heparin sodium (10 IU/mL), antibiotics (100 IU/mL penicillin, 100 μg/mL streptomycin and 0.125 μg amphotericin B. Additional endothelial growth factor (80 μg/mL) was administered. For culture, cells were incubated at 37 °C in humidified air with 5% CO_2_, followed by trypsinization with Trypsin 0.25% (Gibco, Thermo Fisher Scientific, Inc.). HUVECs that had passed from the first to the third passage were placed in gelatin (0.1%) and placed in a 1 mm 48-well tissue culture dish coated with fibronectin (25 μg/mL) and used for cultivation. Cells were identified as endothelial cells by positively labeled acetylated low-concentration lipoproteins and rat anti-human factor VIII. Human polymorphonuclear leukocytes were isolated from the venous blood of healthy adults by standard dextran precipitation and sequential division on Histopaque 1077 (Sigma-Aldrich, St. Louis, MO, USA). Using this method, 95–98% of polymorphonuclear leukocytes were harvested and demonstrated to be 98% pure.

The cell adhesion assay was performed to determine the leukocyte-endothelium interactions using CytoSelect™ Leukocyte-Endothelium Adhesion Assay (Cell Biolabs Inc., San Diego, CA, USA) according to the manufacturer’s protocol. For the experiment, HUVECs were seeded into 96-well plates and cultured with ASA (30, 150 and 300 μg/mL), AGE 232 (30, 150 and 300 μg/mL), or without any drug as the control. A suspension of THP-1 cells (ATCC^®^ TIB202™) at 2 × 10^6^ was prepared in serum-free media and used with the CytoSelect™ kit. After 24 h, HUVEC-treated cell media were aspirated and washed with serum-free media, and 200 μL of THP-1 suspension was added to each well and incubated for 90 min in a cell culture incubator. After 90 min, the plates were washed three times with 250 μL of 1 X wash buffer, and 150 μL of 1 X lysis buffer was added to each well and incubated for 5 min at room temperature. The mixture (100 μL) was transferred to a 96-well plate, and the fluorescence was measured at 520 nm.

### 2.8. COX-½ Enzyme Inhibitor Screening Assay

The COX-½ screening assay was performed to evaluate the ability of AGE 232 and ASA to inhibit COX-1 and COX-2 activities on prostaglandin production by measuring the amount of Prostaglandin 2α produced in the cyclooxygenase reaction. AGE 232 and ASA were dissolved in 100% DMSO to prepare a stock concentration of 10 mg/mL for each sample. AGE 232 and ASA were tested in triplicate at 25, 50 and 100 μg/mL, and concentrations of 1, 2.5, 5 and 10 μg/mL for ASA using a commercial COX inhibitory screening assay kit as recommended by the manufacturer (Cayman test kit-560131, Cayman Chemical Company, Ann Arbor, MI, USA). An aliquot of the mixture was removed, and the prostanoids produced was quantified spectrophotometrically via enzyme immunoassay.

### 2.9. Statistical Analysis

All data are shown as the mean ± SD. The statistical difference between the control and experimental groups was analyzed using a one-way analysis of variance (ANOVA) or a two-sample t-test to determine statistical significance. *P* values of ≤ 0.05, between groups, was considered statistically significant.

## 3. Results

### 3.1. Anti-Thrombotic Effect of Age 232 in a Thromboembolism Mouse Model

AGE 232 significantly reduced death or paralysis caused by collagen/epinephrine-induced thromboembolism in a dose-dependent manner. The protection rates of AGE 232 were 40%, 62.5% and 31.5% at concentrations of 40, 80 and 160 mg/kg, respectively. Decursin, one of the most abundant coumarins present in *A. gigas* Nakai roots, showed a 43.75% protection at a concentration of 12.4 mg/kg. AGE 232 concentrations showed high protection rates compared to those of the control group. Therefore, AGE 232 at 80 mg/kg showed 12.5% higher protection rates than the positive control (Table 1).

### 3.2. Comparison of AGE 232 and ASA in Terms of Toxicity

Through an in vivo study, we found that AGE 232 protected from death or paralysis by collagen/epinephrine-induced thromboembolism. Next, we examined the toxicity of AGE 232 by determining the cell cytotoxicity and ulcer index in the stomach of mice because high doses and long-term treatment with ASA can cause gastric ulcers, compared with ASA. First, we treated AGE232 and ASA to RGM-1 cells with AGE 232 and ASA at various concentrations and analyzed the IC_50_ values

As shown in Figure 1, the MTT assay demonstrated a gradual decrease in RGM-1 cell viability with increasing concentrations of AGE 232 and ASA. In the case of ASA, no significant cellular toxicity was observed at a lower therapeutic dose (0.18–1.8 μg/mL). However, the cellular toxicity was 89% at 36.03 μg/mL (Figure 2A). In the case of AGE 232, the lower doses (10–50 μg/mL) did not show significant cellular toxicity, and the cellular toxicity significantly increased over 100 μg/mL (Figure 2B). Based on these data, IC_50_ values of AGE 232 and ASA were found to be 77.9 μg/mL and 10.2 μg/mL, respectively.

Next, we determined the extent of gastric ulcer as a representative side effect of nonsteroidal anti-inflammatory drugs (NSAIDs) in mice following treatment with AGE 232 and ASA. For this experiment, we used an ASA-induced ulcerogenesis mouse model [17]. First, we measured body weight and feed consumption to check the general body condition. The body weight and food per animal and food intake per cage were recorded daily for five consecutive days at the same time of the day. ASA-treated mice showed a significant reduction in body weight compared with AGE 232-treated mice and the control group (Figure 2A). Mice treated with 300 mg/kg ASA showed a significant decrease in food intake compared to the control group. Mice treated with 30 mg/kg ASA showed a slight decrease in food intake compared with the control group. Mice treated with 300 mg/kg AGE 232 showed a slight decrease in food intake on the second day of treatment and maintained the same intake until the end of the experiment (Figure 2B). We then determined the presence of gastric ulcers in each group. Oral administration of ASA at 300 mg/kg induced significant gastric ulceration compared to the control group (Figure 2C). In the experimental groups, one death was recorded in the ASA-treated group at a concentration of 300 mg/kg on the 4th day of treatment. The stomach mucosa appeared normal in the AGE 232-treated mice compared with the control group. In contrast, ASA-treated mice at a concentration of 30 mg/kg showed smalls hemorrhagic ulcers and some clotted blood was noted in the gastric mucosa. ASA-treated mice with concentrations of 300 mg/kg showed numerous scattered deep hemorrhagic ulcers and clotted blood. Additionally, these mice showed an increase in gas content inside the stomach compared with the control group. (Figure 2D). These results indicate that AGE 232 is a less toxic anti-thrombotic candidate compared to ASA.

### 3.3. Comparison of AGE 232 and ASA on the Mode of Action in Reducing Thromboembolism

Finally, we compared the anti-thromboembolic activity of AGE 232 with ASA. In the case of human platelet aggregation, AGE 232 dramatically inhibited aggregation in a dose-dependent manner compared with the control group. AGE 232 showed aggregation rates of 59 ± 4.9%, 53 ± 6.2%, 43 ± 12.1%, 20 ± 11.5% and 2 ± 0.8% at concentrations of 10, 20, 40, 80 and 160 μg/mL, respectively. ASA showed aggregation rates of 1 ± 0.5% at a concentration of 2.5 μg/mL (Figure 3). In addition, in the adhesion of neutrophils, AGE 232 inhibited the adhesion of neutrophils to HUVECs compared with the control group, but ASA increased the number of neutrophils attached to the HUVECs in a dose-dependent manner (Figure 4). Interestingly, the inhibition of COX-1 and COX-2 activity increased in an ASA concentration-dependent manner (Figure 5A). In contrast, the inhibition of these enzyme activities was decreased by AGE 232 in a concentration-dependent manner (Figure 5B). These results show that AGE 232 has a different mode of action in reducing thromboembolism compared to ASA.

## 4. Discussion

Thrombosis is characterized by clots that affect blood circulation and is the most common cause of death worldwide [18]. Arterial thrombosis is initiated by endothelial injury, which initiates a coagulation cascade and platelet activation [19,20]. Platelet adhesion is an essential process for hemostasis but can initiate adhesion of flowing platelets and initiate an event of firm platelet adhesion, aggregation and thrombus formation [21,22]. Several drugs are used for the treatment and prevention of thrombosis, but most of these drugs have side effects such as bleeding and gastrointestinal injury [13]. Therapeutic approaches using herbal medicines provide an alternative and promising treatment for stroke, with fewer side effects [23].

In the present study, we investigated the effectiveness of AGE 232 as an anti-thrombotic agent in vivo.

The thromboembolism model was modified using DiMinno’s model, wherein paralyzed or dead mice were counted to determine the protective effect of AGE 232 from collagen-and epinephrine-induced pulmonary thrombosis [14]. DiMinno’s model presents no scientific evaluation to judge paralysis in each animal; thus, we decided to use a rotarod test to measure the performance of the treated mice more accurately. This method permits recording the overall performance of the mice in terms of timing, walking, availability of riding and the capacity to maintain balance most accurately and to determine paralysis of animals which is indicated by a limitation in performance, avoiding subjective judgment [24,25,26]. Our results showed, AGE 232-treated mice at concentrations of 40, 80 and 160 mg/kg showed a decrease in the number of paralyzed or dead animals at 40%, 62.5% and 31.25%, respectively, compared with the control group. Decursin-treated mice showed a protection rate of 43.75% compared to the control. The positive group was treated with ASA at a concentration of 100 mg/kg compared with the control group. This result showed that AGE 232 at a concentration of 80 mg/kg resulted in a decrease in paralyzed animals and an increase in protection compared with ASA-treated mice (Table 1).

Antiplatelet drugs principally inhibit the adhesion, activation and aggregation of platelets [27]. The major side effect of antiplatelet drugs is hemorrhage [19]. ASA is an NSAID, and this family of anti-inflammatory drugs causes mucosal damage and ulcer complications principally in the gastrointestinal tract by inhibition of prostaglandin synthesis [28]. ASA irreversibly inhibits cyclooxygenase enzymes. Arachidonic acid can be converted into eicosanoids. Cyclooxygenase enzymes are responsible for the transformation of arachidonic acid to endoperoxides, which are required for the synthesis of prostaglandins, prostacyclins and thromboxanes. [29]. Inhibition of prostaglandin enhances gastric motility, resulting in increased mucosal permeability, neutrophil infiltration and oxyradical production, thereby producing gastric lesions [30]. Long-term treatment with even low doses of ASA, also known as aspirin, induced gastric lesions as a side effect in patients [31].

To determine the cytotoxicity of AGE 232 and ASA, the MTT assay was carried out to check the effect of both compounds on the viability of gastric cells. Aspirin significantly decreased the viability of gastric cells in a dose- and time-dependent manner [32]. As shown in Figure 1, the ASA treatment decreased cell viability in a dose-dependent manner at lower concentrations compared with the AGE 232 treatment. The ASA treatment lowered cell viability at 36.03 μg/mL. However, the AGE 232 treatment decreased cell viability at 500 μg/mL, with similar cellular viability found at concentrations between 150 and 500 μg/mg. The IC_50_ values were measured to compare the cell viability after treatment with both compounds. AGE 232 showed an IC_50_ of 77.9 μg/mL compared with ASA which was 10.2 μg/mL. Thus, the side effects of AGE 232 and ASA on gastric cells in vivo were confirmed by the cell viability results, showing that AGE 232 treatment does not decrease the cell viability compared with ASA treatment at a lower dose. To compare the possible side effects of AGE 232 compared with ASA administration on gastric cells, we used an ASA-induced ulcerogenesis mouse model in which animals were treated with ASA at 30 mg/kg and 300 mg/kg, and AGE 232 at concentrations of 30 mg/kg and 300 mg/kg for five consecutive days. Our results showed that mice treated with ASA at both concentrations showed a decrease in food intake and a decrease in body weight compared with the control group (Figure 1B and Figure 2A). On the day of the sacrifice, the gastric mucosa of low-dose ASA-treated mice showed small ulcers and clotted blood inside the stomach (Figure 2C). ASA-treated mice (300 mg/kg) showed a significant decrease in body weight and food intake compared to the control group (Figure 1B and Figure 2A). On the day of sacrifice, the gastric mucosa showed a high ulcerated area and a high amount of clotting blood and gas inside the stomach, confirming the mucosal side effects related to ASA administration (Figure 2C). However, AGE 232-treated mice at concentrations of 30 and 300 mg/kg did not show significant differences in mucosal damage area compared with the control group (Figure 2D). This result shows that AGE 232 does not have gastric side effects such as ulcerations compared with ASA at the same concentrations (Figure 2).

Arterial thrombosis is initiated by the disruption of the vascular endothelium associated with plaque rupture. In the endothelium, adjacent platelets start to accumulate at the injured site, initiating atherothrombosis [1]. Collagen is one of the most important constituents of the subendothelial matrix and plays an important role in platelet activation [33]. Upon damage to the endothelium, adhesive micromolecules and platelet agonists, such as collagen, von Willebrand factor, adenosine diphosphate and thrombin, are exposed to the damaged site [34]. Collagen activates platelets by transducing signals through glycoprotein VI, which stimulates platelet aggregation using the GPVI-selective agonist convulxin and collagen-related peptide [35]. This activation of the signaling pathway leads to changes in platelet shape and activation of platelet integrins from a low- to a high-affinity state which binds fibrinogen, von Willebrand factor, fibronectin and vitronectin promoting coagulation, and under abnormal conditions can lead to thrombus formation [36].

A previous study suggested that AG coumarins can inhibit collagen-induced platelet aggregation [13]. In the present study, we confirmed that AGE 232 inhibits collagen-induced platelet aggregation in a dose-dependent manner. AGE 232 at a concentration of 160 μg/mL showed a similar antiplatelet aggregation ratio compared with the control and ASA at 2.5 μg/kg showed a high antiplatelet effect compared with the control (Figure 3). An increase in platelet aggregation in the circulatory system activates the inflammatory cascade [37]. The cessation of blood flow induces stress on the vascular endothelium and increases the adhesion of P-selectin to the endothelium. Selectin attracts leukocytes to the endothelial surface and decreases blood flow [38]. Leukocytes initially play a role in the activation of endothelial cells in the inflamed tissue by the interaction between P and E selectin in damaged endothelium. Immunoglobulins mediate the adhesion between leukocytes and endothelial cells, and this activation of leukocytes at the inflammation site leads to activation of the coagulation cascade. Thrombin is the central enzyme in coagulation and is chemotactic to neutrophils and monocytes, stimulating the release of interleukins. Adhesive interactions of leukocytes, endothelial cells and platelets are part of the inflammatory response, starting with the activation of a coagulation cascade, and with this, increasing thrombus formation [39,40]. Previous studies have shown that ASA induces gastritis by promoting neutrophil-endothelial adhesion [41]. Our results showed that ASA increased the percentage of neutrophils attached to the HUVEC cells in a dose-dependent manner; however, AGE 232 showed a decrease in the number of neutrophils attached to the HUVEC cells in a dose-dependent manner, suppressing leukocyte adhesion to the endothelium. This decrease in adhesion can be followed by a decrease in endothelial damage, contributing to the prevention of thrombus formation (Figure 4).

At least two cyclooxygenases (COXs) have been discovered. COX-1 is expressed throughout the body and is particularly important for gastrointestinal protection, and COX-2 is enhanced by cytokines, growth factors and other inflammatory stimuli [28]. COX-1 is expressed in platelets, epithelia of the gastrointestinal tract and other tissues. COX-2 is expressed due to cytokine induction at the inflammation site [42]. COX-1 is an enzyme that converts arachidonic acid to prostaglandin G2 (PGG2), which is converted to PGH2 by peroxidase. In activated platelets, PGH2 is converted to TxA2, which activates new platelets. In endothelial cells, PGH2 is converted to PGI2, an inhibitor of platelet function, by increasing the intracellular levels of cAMP. ASA irreversible acetylates COX-1, blocking TxA2 formation in platelets, which cannot synthesize new proteins, inhibiting their lifespan [19]. Inhibition of COX-1 inhibits prostaglandins, principally prostaglandin E2 (PGE2). PGE2 is produced in the gastric mucosa and has cytoprotective properties. Inhibition of PGE2 synthesis results in gastrointestinal ulcers and patients may also bleed to death [43]. Our results showed that AGE 232 at concentrations of 25, 50 and 100 µg/mL showed a lower inhibition of COX-1 compared with ASA at concentrations of 2.5, 5 and 10 µg/mL in a dose-dependent manner (Figure 5A). This result confirms that AGE 232 administration does not block COX-1 enzyme and PGE2 synthesis involved in bleeding which is found to be a side effect of ASA administration. These results confirm the lower ulceration area or absence of clotting of blood in the stomach (Figure 1D) in AGE 232-treated mice compared with ASA-treated mice. COX-2 is not normally expressed in all the tissues. COX-2 is an enzyme that produces prostaglandins during inflammatory events and is involved in the production of inflammatory mediators, and selective inhibition, and has anti-inflammatory, analgesic and antipyretic effects without gastrointestinal toxicity [44,45]. Our study showed AGE 232 decreased the inhibition of COX-2 ratio in a dose-dependent manner, in contrast to ASA, which increased the COX-2 inhibition in a dose-dependent manner (Figure 5B). AGE 232 at a lower dose (25 μg/mL) showed similar inhibition rates compared with ASA at higher concentrations (10 µg/mL). The result suggests that the antiplatelet effect of AGE 232 is not related to COX inhibition, and more studies are needed to elucidate this pathway.

## 5. Conclusions

In conclusion, AGE 232 has a beneficial antithrombotic effect by suppressing platelet aggregation and the inflammatory cascade, which is the principal component involved in thrombus formation and stroke. AGE 232 treatment also shows low cytotoxicity to gastric cells and no effect on the gastric mucosa in mice such as ulceration or bleeding, the principal side effects of the currently used antithrombotic drugs. In this study, we suggest that AGE 232 may play a beneficial role in improving blood circulation and decreasing thrombus formation, and in the preventive treatment for stroke.

## Figures and Tables

**Figure 1 life-11-00939-f001:**
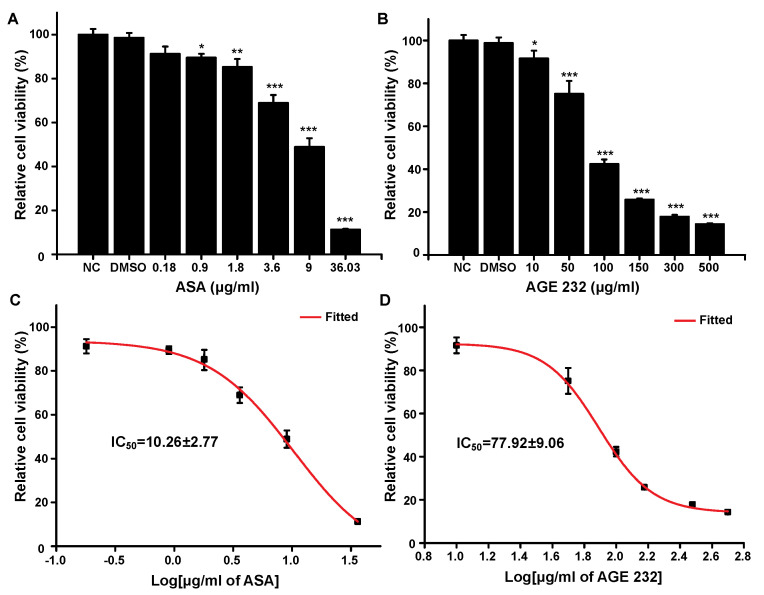
Cell viability assay by MTT. RGM−1 cells (5 × 10^3^) were grown in 96−well plate for 24 h and Table 232. at concentrations of 10, 50, 100, 150, 300 and 500 μg/mL (**A**), and ASA at concentration of 0.18, 0., 1.8, 3.6, 9 and 36.03 μg/mL (**B**) for 24 h. Following the reduction of MTT by metabolically active cells, the formazan crystals formed were solubilized in dimethylsulfoxide and quantified using a microplate reader at 570 nm. IC_50_ curves for RGM-1 cells exposed to AGE 232 (**C**), and ASA (**D**) after 24 h of incubation. Data are expressed as Mean ± SD. * *P* < 0.05, ** *P* < 0.01 and *** *P* < 0.001 compared with the control group. (n = 7). AGE 232, ethanol extract of *A. gigas* Nakai; ASA, acetylsalicylic acid.

**Figure 2 life-11-00939-f002:**
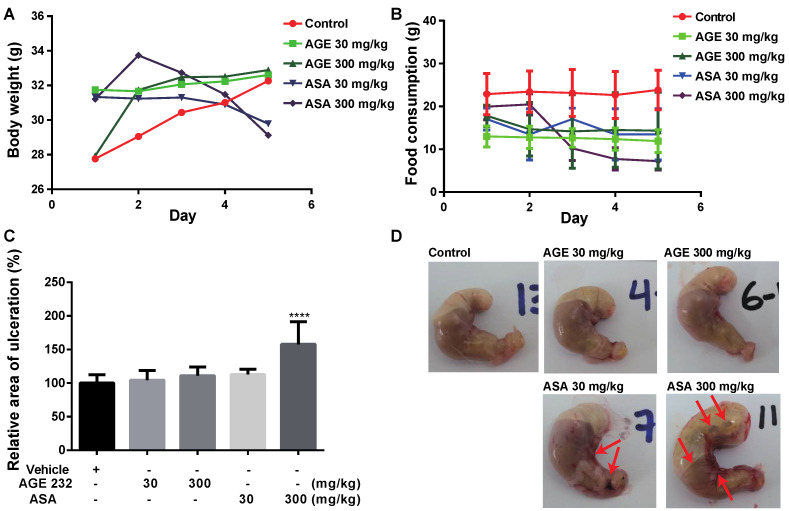
Daily recording of the body weight (**A**) and food intake of mice per cage (**B**) was checked at 8:00 AM every day. The ulcerated area was measure using ImageJ at the end of the 5th day of treatment. The graph shows the ulcerated area in percentage (%) compare with the control at concentrations of 30, and 300 mg/kg of ASA and AGE 232 (**C**). Changes in the stomach of mice treated with AGE 232 at concentrations of 30 mg/kg (2) 300 mg/kg (3), ASA at concentrations of 30 mg/kg (4) and 300 mg/kg (5), compared with control (1). Red arrows indicate the ulcer region in the stomach. (**D**). ASA-treated mice showed congestion. Data are expressed as ± SD. **** *P* < 0.0001 compared with the control group. (n = 10). AGE 232, ethanol extract of *A. gigas* Nakai; ASA, acetylsalicylic acid.

**Figure 3 life-11-00939-f003:**
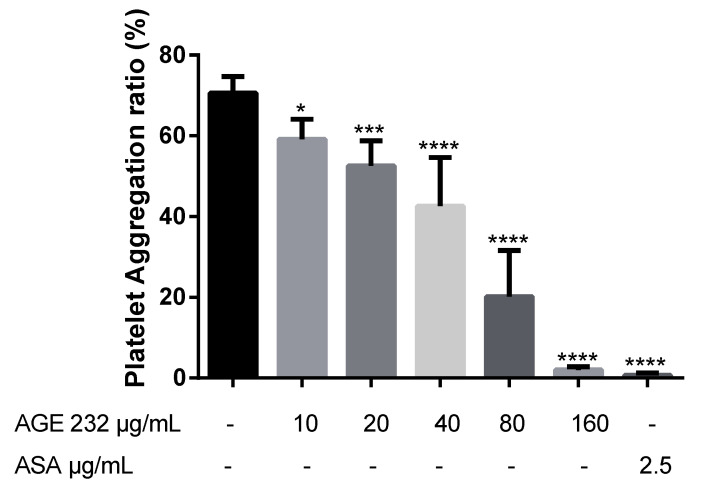
Effect of AGE 232 and ASA on platelet aggregation induced by collagen in human platelet-rich plasma. The graph shows percentage aggregation data of AGE 232 at concentrations of 10, 20, 40, 80 and 160 μg/mL compared with the control, and concentration of 2.5 μg/mL of acetylsalicylic acid (ASA) as the positive control. Data are expressed as ± SD. * *P* < 0.05, *** *P* < 0.001 and **** *P* < 0.0001 compared with the control group (n = 7). AGE 232, ethanol extract of *A. gigas* Nakai; ASA, acetylsalicylic acid.

**Figure 4 life-11-00939-f004:**
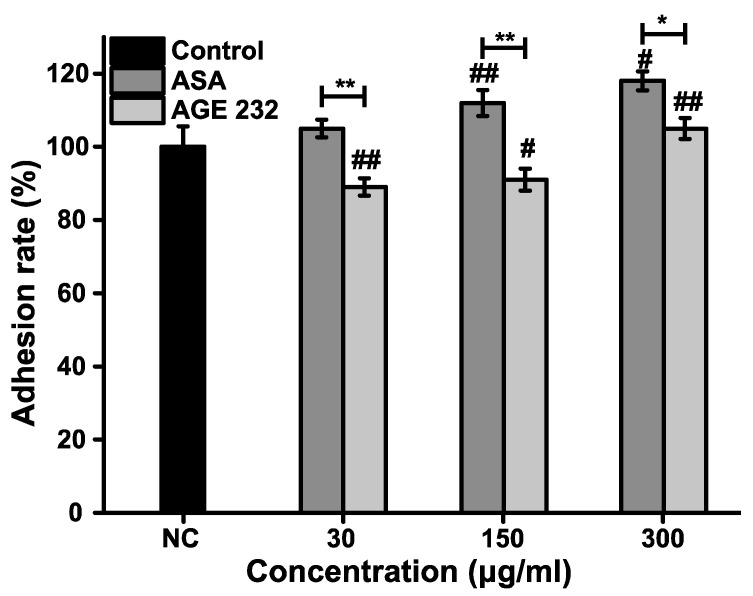
Effect of AGE 232 and ASA in endothelial leukocyte adhesion test. * represents the percentage data of AGE 232 compared with the ASA percentage data. # represents the percentage data of AGE 232 compared with the control group. Data are expressed as ± SD. * *P* < 0.05, ** *P* < 0.01, # *P* < 0.05, ## *P* < 0.01 compared with the control group. (n = 4). AGE 232, ethanol extract of *A. gigas* Nakai; ASA, acetylsalicylic acid.

**Figure 5 life-11-00939-f005:**
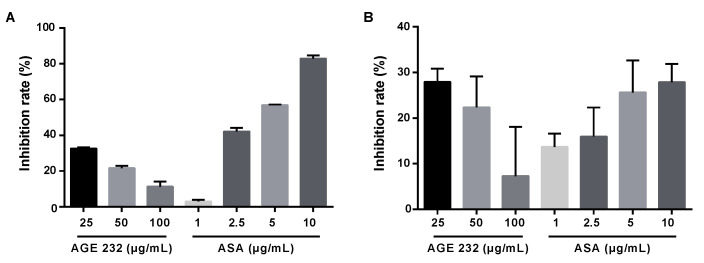
Inhibitory effect of AGE 232 at concentrations of 25, 50 and 100 µg/mL, and ASA at concentrations of 1, 2.5, 5 and 10 µg/mL of COX-1 (**A**), and COX-2 (**B**). Data are expressed as inhibition rates (%). AGE 232, ethanol extract of *A. gigas* Nakai; ASA, acetylsalicylic acid. COX-1 and COX-2 are cyclooxygenases.

**Table 1 life-11-00939-t001:** Protective effect of AGE 232 on the thromboembolism mouse model.

Group	Dose (mg/kg)	Number of Dead or Paralyzed Mice/No. of Total Mice	Protection (%) *
Control	Vehicle	13/15	13.13
ASA	100	8/16	50.0
AGE 232	40	9/15	40.0
	80	6/16	62.5
	160	9/16	31.25
Decursin	12.4	9/16	43.75

* Protection (%) indicates the percentage of live and non-paralyzed mice in each group. AGE 232, ethanol extract of *A. gigas* Nakai; ASA, acetylsalicylic acid.

## Data Availability

The data presented in this study are available in the article.

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
