# Peer review of "Antithrombotic Effect of the Ethanol Extract of Angelica gigas Nakai (AGE 232)"

_life, 2021, doi:10.3390/life11090939_

Round 1
Reviewer 1 Report
Overall, this is a well written paper. The research design is appropriate, the methods are extensively described and the results are interesting. Minor spell check would be required.
Author Response
Response to Reviewer 1 Comments
Reviewer's point
Overall, this is a well written paper. The research design is appropriate, the methods are extensively described and the results are interesting. Minor spell check would be required.
Response
First of all, I appreciate your revision. I fixed something according to your comment like in revised manuscript like below.
Line 38-39: intracerebral hemorrhage (10% of total cases), subarachnoid hemorrgage (3% of the cases), and ischemic (87% of the total cases) ® subarachnoid hemorrhage, intracerebral hemorrhage, and ischemic stroke (3%, 10%, and 87% of the total cases respectively)
Line 51: artery and → artery, and
Line 60: most commonly used → most used
Line 86: 5 week-old → 5-week-old
Line 222: was considered to be → was considered
Line 328: hat affect blood circulation, and → that affect blood circulation and

Reviewer 2 Report
The manuscript presents data on the elucidation of antithrombotic effect of the ethanol extract of Angelica gigas 2 Nakai (AGE 2), traditional herbal medicine used in the treatment of various illnesses. In the presented study the antithrombotic effect of the ethanol extract of A. gigas Nakai was evaluated in the in vivo and in vitro models.
The rationale for the establishment of this work ascertains its novelty and the importance in the context of the medicinal properties of tested plant and potential application in drugs of the plant origin and the work can be recommended for publication in the Life journal.
Author Response
Reviewer's point
The manuscript presents data on the elucidation of antithrombotic effect of the ethanol extract of Angelica gigas 2 Nakai (AGE 2), traditional herbal medicine used in the treatment of various illnesses. In the presented study the antithrombotic effect of the ethanol extract of A. gigas Nakai was evaluated in the in vivo and in vitro models.
The rationale for the establishment of this work ascertains its novelty and the importance in the context of the medicinal properties of tested plant and potential application in drugs of the plant origin and the work can be recommended for publication in the Life journal
Response
I appreciate your revision.